# Machine Learning Applications and Optimization of Clustering Methods Improve the Selection of Descriptors in Blackberry Germplasm Banks

**DOI:** 10.3390/plants10020247

**Published:** 2021-01-28

**Authors:** Juan Camilo Henao-Rojas, María Gladis Rosero-Alpala, Carolina Ortiz-Muñoz, Carlos Enrique Velásquez-Arroyo, William Alfonso Leon-Rueda, Joaquín Guillermo Ramírez-Gil

**Affiliations:** 1Corporación Colombiana de Investigación Agropecuaria—AGROSAVIA, Centro de Investigación La Selva- Km 7, 250047 Ríonegro, Colombia; jhenao@agrosavia.co (J.C.H.-R.); mroseroa@agrosavia.co (M.G.R.-A.); cortizm@agrosavia.co (C.O.-M.); cvelasquez@agrosavia.co (C.E.V.-A.); 2Departamento de Agronomía, Facultad de Ciencias Agrarias, Universidad Nacional de Colombia, 111321 Sede Bogotá, Colombia; waleonr@unal.edu.co

**Keywords:** random forest, K-means, morphological descriptors, digital agriculture, data science

## Abstract

Machine learning (ML) and its multiple applications have comparative advantages for improving the interpretation of knowledge on different agricultural processes. However, there are challenges that impede proper usage, as can be seen in phenotypic characterizations of germplasm banks. The objective of this research was to test and optimize different analysis methods based on ML for the prioritization and selection of morphological descriptors of *Rubus* spp. 55 descriptors were evaluated in 26 genotypes and the weight of each one and its ability to discriminating capacity was determined. ML methods as random forest (RF), support vector machines, in the linear and radial forms, and neural networks were optimized and compared. Subsequently, the results were validated with two discriminating methods and their variants: hierarchical agglomerative clustering and K-means. The results indicated that RF presented the highest accuracy (0.768) of the methods evaluated, selecting 11 descriptors based on the purity (Gini index), importance, number of connected trees, and significance (*p* value < 0.05). Additionally, K-means method with optimized descriptors based on RF had greater discriminating power on *Rubus* spp., accessions according to evaluated statistics. This study presents one application of ML for the optimization of specific morphological variables for plant germplasm bank characterization.

## 1. Introduction

Machine learning (ML) is a form of artificial intelligence (AI) that gives machines the ability to learn through the use of algorithms and a training process [1] and is used in tandem with big data technologies and high-performance computing [2,3], which, together with information and communication technologies (ICTs) and the Internet of Things (IoT), deep learning, among others tools, have created new opportunities for data-intensive science. These tools are being applied in multiple areas, including agriculture, as an emerging technology [2,4]. These tools helped create what is now known as digital agriculture, a new agricultural revolution [2,5].

The practical applications of ML in agriculture are broad and are applied in various fields, such as yield prediction [6], pest detection, classification, monitoring, and management [3,7], and species recognition [8], which is why ML is undoubtedly a powerful tool within digital agriculture, providing information for correct decision-making including the study and efficient use of tropical genetic resources. The ML approach in agriculture presents many challenges, because it is highest impacted by environmental factors that cannot be controlled, demanding rigorous process and extensive data for validation and testing [9]. This implies avoiding the overfitting, select very complex models and data without standardization, improve the statists used, and data security, among others [2,9,10]. The goal of good practices in ML could improve real applications in agriculture with biological, ecological, agronomic, economic, cultural, and social contexts.

The subcategories of ML algorithms present different approaches, with variations in mathematical and statistical principles, computational requirements, advantages and disadvantages, and current and potential applications [9,10]. The most common algorithms associated with ML such as random forests, decision trees, supper vector machine, Bayesian networks, neuronal networks, regression analysis, among others are named supervised learning [9]. On the other hand, when unlabeled datasets without prior knowledge of the input and output variable response are named unsupervised learning such as Artificial Neural Networks, genetic algorithm, Instance-based learning models, deep learning, and clustering [2,7,9,10].

Classification is one of the most common processes in which ML algorithms are used. For this purpose, decision trees are the most popular tools, highlighting the random forest algorithm [2,10]. On the other hand, supper vector machine and neuronal networks are algorithms that have been used for this objective [2,9]. In addition, in recent years deep learning has gained popularity in the classification processes of biological phenomena [3,7].

Colombia is one of the more biodiverse countries on the planet because it is rich in natural resources and has a variety of species and ecosystems [11]. In order to conserve and preserve high genetic diversity in species of agricultural interest, the System of Germplasm Banks of the Nation for Food and Agriculture (SBGNAA) was created by the Colombian Agricultural Research Corporation, AGROSAVIA. Banks include collections of species of high value for the nation as a result of their potential for agribusiness, pharmaceuticals, textiles, and food security and sovereignty [12,13]. The SBGNAA genetic pool includes an ex situ blackberry collection (*Rubus* spp.) that is made up of different species with high potential for the domestic and export markets given their nutritional and functional characteristics [14,15].

The value of conserved plant genetic resources is given by the information used to promote their use, where morphological and genetic descriptors are the basic parameters that contribute to the knowledge of variability and parental selection, providing useful information for the development of new cultivars and the protection of varieties [16]. Morphological characterization is the determination of a set of phenotypic traits measurement based on key botanical-taxonomic named as descriptors. The descriptors should have a highly polymorphic expression, stable in the population and expressible under different environmental conditions [17].

Currently, the Colombian *Rubus* spp. collection does not have validated descriptors. Therefore, a series of morphological, chemical, and physiological descriptors is needed in order to provide easy-to-measure characteristics that relate the shape, structure or functions of an accession, allowing compliance with distinguishability, uniformity, and stability requirements [18].

After a proper validation process for descriptors in germplasm banks, the phenotypic variability classification stage continues, which has traditionally been based on multivariate analysis, commonly using principal components, clustering methods such as cluster analysis, and linear and quadratic discriminants [19]. These tools have been used to recognize cultivated and wild species of *Rubus* spp. and commercial crops of *Rubus glaucus* Benth [20,21], without a prior optimization and validation process for descriptors or evaluating which discrimination method had the best performance.

Therefore, given the need to have morphological descriptors for the *Rubus* spp. germplasm bank along with the lack of knowledge on the ability of other discrimination methods to provide specific information on possible descriptors, machine learning tools, and discrimination method optimization have provided an opportunity for the creation of more efficient processes in terms of information capacity, computational performance and optimization of work time in the field and laboratory. The present study aimed to evaluate different machine learning techniques for the selection of morphological variables in 26 *Rubus* spp. accessions cultivated under tropical conditions and for the optimization of discrimination methods to increase the differentiation capacity of accessions.

## 2. Materials and Methods

### 2.1. Rubus spp. Accession Genotypes and Agronomic Management

A total of 26 blackberry accessions (*Rubus* spp.) were selected from the germplasm bank at the La Selva Research Center in Rionegro, Antioquia—Colombia located at an altitude of 2100 masl with an average temperature of 16 °C and 74.83% relative humidity (Longitude: 075°24′51.9 and Latitude: 06°07′52.7), a lower montane humid forest life zone (bh-MB).

The selected accessions had interspecific morphological variation at the species level, obtained as a representative sample of all variation of the *Rubus* spp. germplasm bank collection. The origin of each accession was as follows: Cultivated natives (CN) with 18 accessions; introduced (I) with 6 accessions; and wild native (WN) with 2 accessions (Table 1). The plant material was collected in the phenological phases of flowering; the primary floriferous stems were obtained after renewal pruning to guarantee their quality. For each accession, five plants were selected, which had three floriferous primary stems extracted from each one. The sanitary and renewal pruning were standard for all accessions, as well as the fertilization practices. When there was a water deficit, water was added with automatic drip irrigation.

### 2.2. Morphological Descriptors

The selected morphological descriptors were based on previous studies on morphological characterizations carried out by Evans and Weber [22], combined with the proposals of Ligarreto-Moreno, Espinosa, Barrero and Medina [21] for species from the *Rubus* L genus. Once selected, they were grouped using typology criteria, botanical terminology, and measurement scales. These descriptors were verified with taxonomic keys of the genus *Rubus* L. [20] and the characters for the distinction of varieties of blackberry [21]. This validation confirms that morphological traits are associated with stable phenotypic expressions into *Rubus* genus and low environmental effect.

The descriptors were recorded at the plant and organ levels, including qualitative variables such as growth habit, organ shape, and other vegetative structures. In addition, the organs were colored using the Royal Horticultural Society’s color chart for plants [23]. The quantitative variables focused on morphometric measurements such as length (linear dimension) and length and width (horizontal dimensions). Discrete variables were also used for the vegetative structures in the plant organs (Table 2).

### 2.3. Selection and Optimization of Morphological Descriptors from the Rubus spp. Germplasm Bank Using Machine Learning Tools

The *Rubus* spp. germplasm bank currently does not have validated morpho-agronomic descriptors, meaning much methodological effort is needed for characterization, which is why three ML tools were used to discriminate and determine the weight of the descriptors in terms of diversity for optimal clustering of the evaluated accessions. This study used an internal optimization process from ML algorithms random forest (RF), linear (SVMl) and radial (SVMr) support vector machines, and neural networks (NN).

The data matrix was associated with 55 columns (morphological descriptors) per 78 rows (26 accessions with 3 replicates by each one). In addition, one-hot encode data was realized through a numerical representation of categorical descriptors. The arrays data were randomly divided into two data sets: (i) training (75%) and (ii) testing (25%) to be evaluated under different ML methods.

The RF algorithm is widely used as a classifier given its simplicity in terms of the parameters required for decision making [24]. It was implemented using the libraries randomForest [25], caret [26], ranger [27], and h2o [28] available to free software R.

The optimization of the RF algorithm and determination of the importance of the morphological descriptors (numerical and non-numerical) were carried out through a multi-step analysis. The first step was evaluating the number of trees (set of combinations from 100 to 1000), the size through the number of nodes (1 to 4000), and the hyperparameter alpha (0 to 20) with of the error rate, Bayes error and out-of-bag root-mean-square error (OOB-RMSE), using the caret, ranger and h2o libraries, selecting 500 trees, 4000 nodes and an alpha of 4 as a balance between the robustness, stabilization of the error rate and computational performance. Subsequently, the importance of the morphological descriptors was determined by calculating the confusion matrices [29] and then their global accuracy [30] using the “caret” package. These parameters identified the weight and interactions within the decision tree for each variable. As a complement, the importance of the descriptors was evaluated by calculating the Mean Decreases in the Gini Index metric. This process was corroborated using the metrics: mean minimal depth, number of nodes, mse increase, node purity increase, number of trees times a root, root variable, interaction occurrences, uncond mean minimal depth and significance (*p* < 0.05), evaluated with the randomForestExplainer library [31]. Finally, these metrics were graphed. The type of assembly used in the RF algorithm for the selection of the models was Bagging.

The second evaluated algorithm was SVM, which belongs to the general category of kernel methods, widely used in classification and regression because of its high precision and capacity to handle high-dimensional data [32]. The linear and radial forms of the SVM algorithm (SVMl and SVMr) were evaluated using the e1071 library [33], optimized with two steps: (i) selection of parameters and (ii) final training [34]. To select the fitness parameters, the classification error, and the mean square error of the regression were determined using the e1071 library. In the case of SVMr, gammas of 0.5, 1, 2, 3, 4, and 5 were tested, and the computational cost was determined with the sum of the hyperparameters and an indirect measurement of the computational simplicity of the code.

The third algorithm was NN, commonly used in identifying and predicting patterns between multiple variables [35]. It was implemented with the nnet [36], NeuronalNetTools, and RSNNS packages [37]. The importance and sensitivity of the descriptors were evaluated using the Garson and Olden algorithms and the Lek profile method; then, the network was optimized through a step-by-step process [35]: (i) Normalize entries, standardize responses, and evaluate the influence of outliers. (ii) Network architecture, which includes the size or number of units in the hidden layer, the number of nodes in each layer, inclusion of bias layers, and weights or inputs. (iii) Decay by decreasing the specific weight of the regularization in the neural network and (iv) interactions by evaluating different amounts of interactions. Additionally, the correlation matrix between variables was calculated [35].

With the optimization of each algorithm (RF, SVMr, SVMl, and NN), the results obtained in the selection of descriptors and its ability to classify adequately the accessions of *Rubus* spp. were evaluated based on training/validation accuracy and training/validation missclass Error using area under receiver operating characteristic (ROC) curve (AUC) [38] implemented in the free software R [39] with own code. The accuracy quantified by AUC it is considered a good metric that has been used in the comparison of ML algorithms [40].

### 2.4. Rubus spp. Germplasm Bank k Genotype Clustering Methods

The tools selected for the analysis of discrimination of the accessions in the *Rubus* spp. germplasm bank included hierarchical agglomerative clustering, using the Ward D2 method as the clustering union strategy, and k-means [41]. The analyses were developed in the R software [39] through the creation of an own code with the help of the libraries vegan [42], pvclust [43], ape [44], and rgl [45]. The consolidated morphological descriptors in Table 2 were used as discriminant variables.

In order to optimize the discrimination of genotypes based on morphological descriptors, a multistage analysis was carried out: (i) Standardization of variables carried out by means of the Z score using the clusterSim library [46], guaranteeing equal or similar measurement scales especially in measurements of dissimilarity sensitive to magnitude such as the Euclidean distance [47,48]. (ii) For both discrimination methods, the optimal number of clusters was estimated with the gap statistic [49] using the libraries factorextra [50] and stat [51]. (iii) Since there are no validated morphological descriptors for the germplasm bank, the effect of the number of variables in the discrimination methods was determined using two data sets: (a) all variables (Table 2) and (b) those selected using the RF algorithm (V23, V44, V24, V49, V25, V27, V50, V29, V28, and V42).

The variations in the hierarchical agglomerative clustering and k-means methods were evaluated using the statistics normalized variation index (NVI), Adjusted Rand index (ARI), separation index (IS), Calinski-Harabasz index (CH), entropy (EN), and Pearson Gamma (PG) [52]. The parameters were determined with the fpc v2.2-5 [53] and aricode v0.1.2 libraries [54], implemented in the free R software [39].

All combinations of morphological descriptors used on discriminating methods included the results of ML algorithms were evaluated for their ability to discriminate each accession used as replicas in order to avoid the artifacts such as the environmental condition in differential expression of morphological markers. In addition, the results were taxonomically corroborated to detect anomalies in the algorithms evaluated.

## 3. Results

### 3.1. Selection and Optimization of Machine Learning Algorithms for the Prioritization and Selection of Morphological Descriptors in Rubus spp.

Table 3 shows the results of the ability of each of the machine learning methods to prioritize the importance of morphological variables and their discriminating ability in the studied blackberry accessions after an internal optimization process. It was determined that the RF algorithm had the best performance based on the test statistics. In decreasing order of ability to discriminate adequately based on the appropriate selection of descriptors, the algorithms were RF (descriptive and numerical variables), RF (numerical variables), neuronal networks (NN), support vector machine (SVM) radial (SVMr) and linear (SVMl) with area under curve (AUC) accuracy classification values of 0.76, 0.64, 0.31, 0.21, and 0.09, respectively.

The RF algorithm, after optimization based on the number of trees, and size and reduction of the hyperparameters, was able to determine that the quantitative and qualitative descriptors of greatest importance for use in the discrimination of blackberry genotypes were those with the highest value for the mean decrease in the Gini index (Figure 1a and Figure 2a), minimum depth within the decision forest (Figure 1b and Figure 2b), maximum number of connected nodes (Figure 1c and Figure 2c), increase in purity (Figure 1d and Figure 2d) and number of most frequent interactions in the decision trees (Figure 1e and Figure 2e). According to these criteria, the variables in decreasing order of importance for quantitative and quantitative-descriptive descriptors were V44, V24, V42, V49, V25, V29, V50, V27, V28, V26 and V23, V44, V9, V24, V53, V22, V25, V49, V28, V50, and V26 (Figure 2a–e).

The optimized SVMl, varying parameter C, as a balance between the massification of the algorithm margin and the error with a selection factor of 0.04, determined that the descriptors in increasing order of importance were V51, V29, V19, V7, V27, V32, V46, V31, and V23 (Figure 3a,b). The SVMr found that the selection factor 50 in the Kernel function and a gamma of 0.5 minimized the error at the lowest possible computational cost, which prioritized the descriptors in decreasing order as V19, V37, V46, V48, V23, V30, V42, V44, V51, V7, V32 and V31 (Figure 3b).

NN found that the best relationship as a function of the decrease in the specific weight of the regularization of the network with respect to the number of hidden layers quantified using the Bootstrap indicator determined that the highest value of occurrences was seen with a weight of decay of 0.1 and a number of layers of 5.0 (Figure 4b). The relationship between the variables and the network determined using Olsen’s connection weights algorithm showed that the descriptors with importance greater than 2.5 in absolute value were V30, V34, V32, V31, V49, V12, V50, V26, V33, V36, V48 and V19 (Figure 4a). Additionally, as a result of the correlation analysis, two groups were found: (i) directly proportional relationships and (ii) inversely proportional relationships (Figure 4c). The descriptors associated with the flower, such as V49 and V50 (inversely proportional), helped differentiate the accessions at the inter- and intraspecific levels. The leaf descriptors, such as V31 and V32 (directly proportional), were able to discriminate using a single descriptor. Grouping also occurred in five clusters associated with the importance value of the variables: High importance: V34 to V44, medium importance: V26 to V33, and low importance V12 to V25. The importance of variables V27, V37, V43, and V44 was notable in all groups (Figure 4d).

### 3.2. Genotype Discriminating Methods from the Rubus spp. Germplasm Bank

It was found that the massification of the Gap statistic was obtained when the number of clusters for the standardized hierarchical agglomerative clustering, non-standardized hierarchical agglomerative clustering, standardized k-means, and non-standardized k-means methods were 10, 6, 10, and 7, respectively (Figure 5), indicating a uniform, non-random distribution of the accessions within each group. Additionally, the effect of standardization on the two methods tested was notable (Table 4). Superior performance was found in the K-means and hierarchical agglomerative clustering grouping when the reduction of variables was carried out using the RF method, indicating that the selection process was highly informative (Figure 6 and Table 4).

The evaluation of the behavior and discrimination capacity of the hierarchical agglomerative clustering and k-means methods with their different evaluated variations based on the Normalized Variation Index (NVI); Adjusted Rand Index (ARI); Separation index (IS); Calinski-Harabasz Index (CH); Entropy (EN); Pearson Gamma (PG) indices showed that the method that presented the best performance was K-means with standardized and reduced variables as a function of the RF optimization process, followed by the K-means method with all standardized variables, standardized agglomerative hierarchical grouping with optimization of variables using RF, agglomerative hierarchical grouping with all standardized variables, K-means and non-standardized agglomerative hierarchical grouping (Table 4).

Figure 7 shows the relationship between the descriptors prioritized by the RF algorithm and the results of the discrimination using the K-means method, without contradictory variations in the discriminating morphological characteristics for each group or accession.

## 4. Discussion

### 4.1. Selection and Discrimination of Descriptors Applied to the Rubus spp. Germplasm Bank Using Machine Learning Tools

Of the analyzed machine learning tools, RF presented the best performance when compared to SVMl, SVMr, and NN for the ability to prioritize highly discriminating descriptors of accessions from the *Rubus* genus. This algorithm has been widely used in classification processes given its good behavior, simplicity in terms of requirements and parameters, computational optimization capacity, high precision, and robustness to noise, among other reasons [55]. The classification values (AUC) found in this study agree with similar processes where the RF algorithm was used to determine classes associated with different phenotypes and landscape uses with information from remote sensors, among other uses [56].

Machine learning is one of the most used tools today in various fields, including agriculture [2,57], which has established itself as one of the most effective methods for detecting and predicting patterns [58]. Generalized use poses a challenge in the user community for adequate applications; an optimization and analysis process is needed for each algorithm, such as the one developed in this study.

The biological interpretation of the optimization and selection of morphological descriptors in the *Rubus* spp germplasm bank based on RF algorithm were variables and informative into *Rubus* genus, maximizing the contrast in the phenotypic discrimination [59]. These results suggest that some numerical morphometric characteristics largely define the interspecific morphological variability of the evaluated *Rubus* accessions, especially those related to the size of the plant organs.

The numerical descriptors with the highest level of discrimination, such as V26 (number of stingers in the stem internode) and V24 (length of the base of the stinger in the stem), are characteristics related to V23 (shape of the stinger in the stem), which makes variations in this vegetative structure a very valuable indicator that easily discriminates accessions or materials. Descriptors related to the size and arrangement of the leaf on the stem were also very important. Since species are represented by accessions and intraspecific variability, our results generally confirmed that the descriptors associated with the leaf and stem (Figure 7) tend to be the most informative [60]. Studies on *Rubus* subgenus *Rubus* highlight these descriptors in the determination of qualitative and quantitative variation among accessions [60]. These results suggest that many of the numerical descriptors prioritized by the RF method show the possibility of generating scale ratios, which is useful for comparing intervals, differences, and derivatives in absolute or dimensionless values [60]. Therefore, prioritizing these descriptors would allow the generation of composite indicators and more informative comparisons between collections and plant germplasm banks from different regions [61].

Likewise, when descriptive variables were incorporated both interspecific and intraspecific characteristics helped discriminate the phenotype of the *Rubus* genus materials. With this combination, the descriptors V9 (primary color of the stem surface), V23 (shape of the stinger in the stem), and V53 (primary color of the petals) contributed greatly to the definition of the morphological descriptors for this genus, highlighting the importance of the stinger shape characteristic in the stem, which has proven to be a highly discriminating characteristic at the species level and between species [21].

The 11 phenotypic variables allowed the discrimination of accessions, considered as the minimum morphological characters that would facilitate the study of *Rubus* germplasm. This optimization will allow characterizing the phenotypic traits of the accessions and covering a large number in a short time, reducing the time for the characterization of the entire germplasm bank [17].

### 4.2. Selection of the Genotype Clustering Method of the Rubus spp. Germplasm Bank

In the hierarchical agglomerative clustering and K-means, the standardization process combined with the determination of the optimal number of clusters presented a better discriminant power for *Rubus* spp. accessions. If the initial population and its distribution have large distances between individuals, produced by the use of non-standardized values, the number of clusters produced by the Gap statistic tends to be low and, therefore, the discrimination power is lower [49]. This affects the K-means and hierarchical agglomerative clustering methods since the Gap calculation includes the logarithm of the Sum of Square Errors in modal distribution in all terms as a quotient [62].

The NVI, ARI, CH, and PG statistics indicated that the greatest effectiveness was seen with the standardized K-means variants with reduced variables, standardized K- means with all variables, and standardized hierarchical agglomerative clustering with all the variables, because said statistics related the number of classes, their internal normalized variations and the order of the scores between the different classes formed. Therefore, they were strongly influenced by the number of clusters, the distances between individuals, and the nature of the grouping methods [63].

The K-means method performed better than hierarchical agglomerative clustering. This result was due to the reassignment nature of the K-means method, which allows each permutation to have an individual assigned to a group, independent of the group it was assigned to in the immediately previous permutation, contrary to the hierarchical methods, where individuals are assigned to a cluster depending on the initial parameters and remain in that group until the end of the analysis, creating subgroups in lower hierarchies [64].

This study indicated that the K-means method and the reduction of variables using the RF algorithm are an excellent alternative for the descriptors optimization and discrimination of accessions from the *Rubus* spp. germplasm bank, with high potential for use in fast, efficient characterizations in other germplasm banks. Even with the promising results presented here, these methods require internal validation, proper selection of the combination of variables, and specific clustering models for replication in different plant matrices [65].

The knowledge of the morphological variability of germplasm improves the understanding of the relationship between structural morphology and their corresponding functional botany [66]. It is considered that in the case of financial or human resource limitations, less relevant characters can be eliminated with objective elements, such as the process performed in this work. In addition, morphological descriptors must be easily determined and have a constant phenotypic expression in all environments—that is, high heritability and low environmental influence. Optimization of descriptors improves the availability of information quickly and accurately inducing efficient management of conservation and maximizing the use of financial resources [13]. Based on previous assumptions, our work constitutes an important contribution in the evaluation of the morphological variation of the *Rubus* germplasm with statistical, botanical, and taxonomic validity.

## 5. Conclusions

The correct optimization process of the RF algorithm allowed stable morphological descriptors with high taxonomic concordance to be selected, thus eliminating redundant and obsolete descriptors that present a high cost–benefit ratio. The adequate combination of discriminant morphological descriptors in combination with the optimal parameters of the K-means clustering method showed a promising approach to discriminate different materials from a population with high phenotypic variability, such as the *Rubus* spp. germplasm bank. This is particularly valuable since it is the first report on the use of machine learning tools and optimization of discriminant methods for the prioritization of quantitative and qualitative morphological descriptors and the ability to differentiate genotypes from plant germplasm banks for the *Rubus* spp. genus in tropical environments.

## Figures and Tables

**Figure 1 plants-10-00247-f001:**
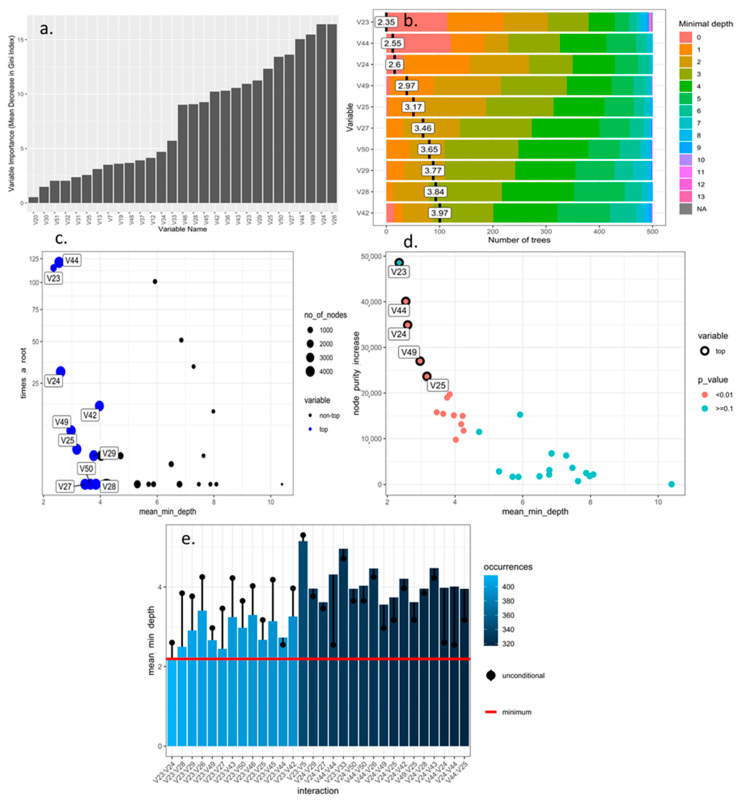
Optimization of numerical morphological descriptors of blackberry (*Rubus* spp.) using machine learning Random Forest algorithm. (**a**) importance of variables based on purity of classification nodes using Gini Index; (**b**) importance of variable based on minimum depth locations between of tree and number of trees; (**c**) classification of top and not variables based on minimum average depth, number of nodes and times to root; (**d**) significance (*p* < 0.05) based on increase in the purity of the nodes and mean minimum depth; (**e**) mean minimum depth for the 30 most frequent interactions variables.

**Figure 2 plants-10-00247-f002:**
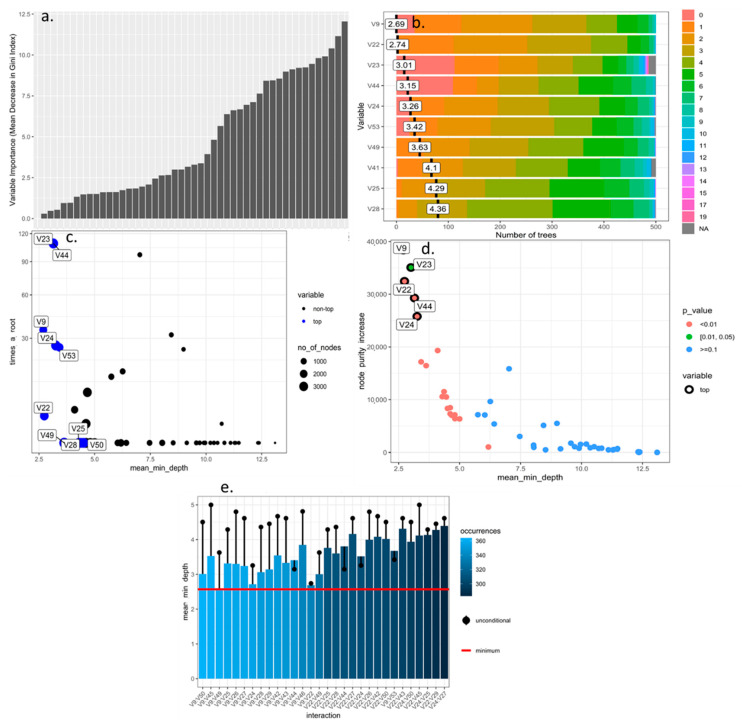
Optimization of all morphological descriptors (numerical and categorical) of blackberry (*Rubus* spp.) using machine learning Random Forest algorithm. (**a**) importance of variables based on purity of classification nodes using Gini Index; (**b**) importance of variable based on minimum depth locations between of tree and number of trees; (**c**) classification of top and not variables based on minimum average depth, number of nodes and times to root; (**d**) significance (*p* < 0.05) based on increase in the purity of the nodes and mean minimum depth; (**e**) mean minimum depth for the 30 most frequent interactions variables.

**Figure 3 plants-10-00247-f003:**
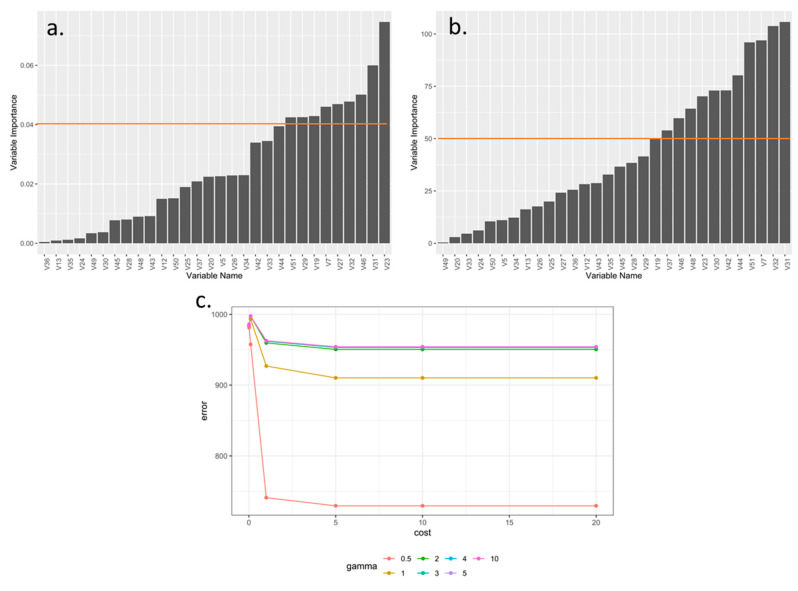
Optimization of morphological descriptors of blackberry (*Rubus* spp.) using machine learning support vector machine algorithms. (**a**) variable relevance based on support vector machine form linear algorithm; (**b**) variable relevance based on support vector machine form radial; (**c**) Parameter optimization associate with support vector machine form radial.

**Figure 4 plants-10-00247-f004:**
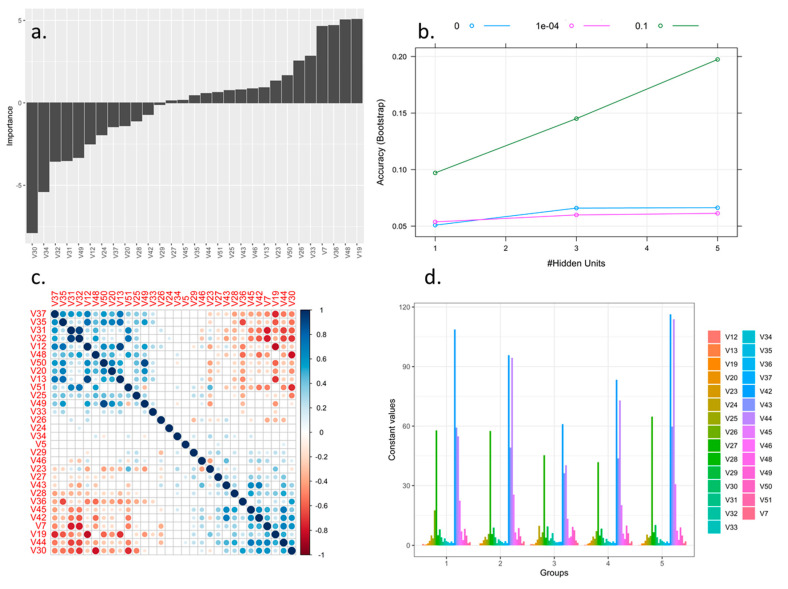
Optimization of morphological descriptors of blackberry (*Rubus* spp.) using machine learning Neuronal Network algorithms. (**a**) Variable importance for models using Olden’s connection weights algorithm.; (**b**) calibration validation and prediction optimizing neural network parameters through Bootstrap; (**c**) Correlogram of interaction of predictive variables in the neuron network; (**d**) Cluster centers for each variable in each group.

**Figure 5 plants-10-00247-f005:**
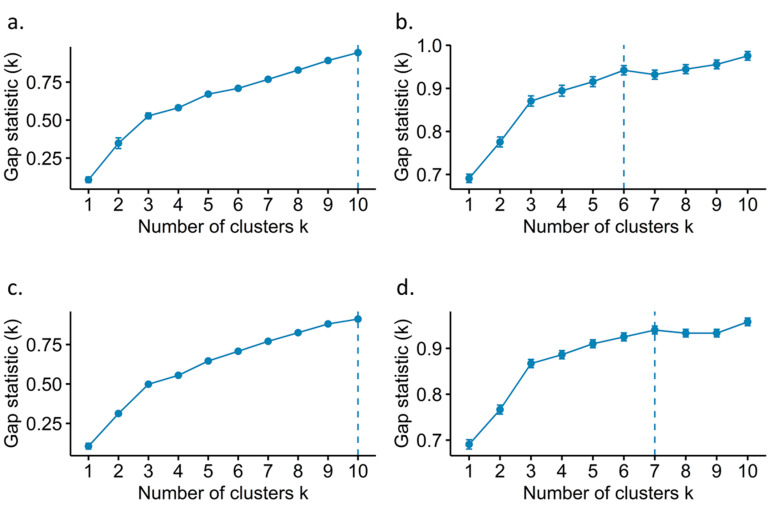
Cluster number selection based on GAP statistics. (**a**) Hierarchical agglomerative clustering (standardized); (**b**) Hierarchical agglomerative clustering (not standardized); (**c**) K-means (standardized); (**d**) K-means (not standardized).

**Figure 6 plants-10-00247-f006:**
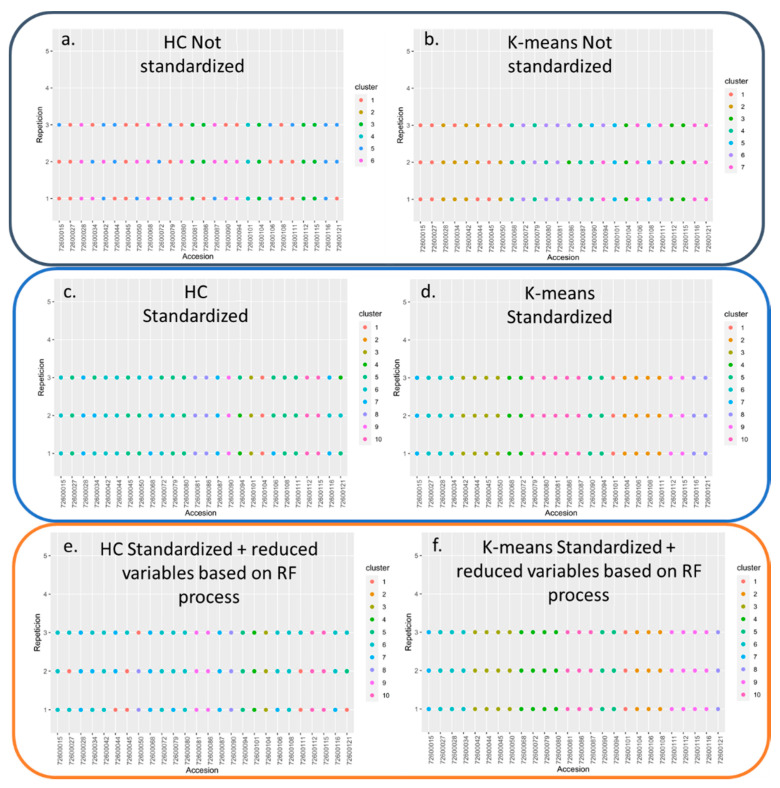
Selection of the genotype clustering method of the **blackberry** (*Rubus* spp.) germplasm bank. (**a**) Hierarchical agglomerative clustering (not standardized); (**b**) K-means (not standardized); (**c**) Hierarchical agglomerative clustering (standardized); (**d**) K-means (standardized) (**e**) Hierarchical agglomerative clustering (standardized and with reduced variables based on Random Forest process); (**f**) K-means (standardized and with reduced variables based on Random Forest process). HC: Hierarchical agglomerative clustering.

**Figure 7 plants-10-00247-f007:**
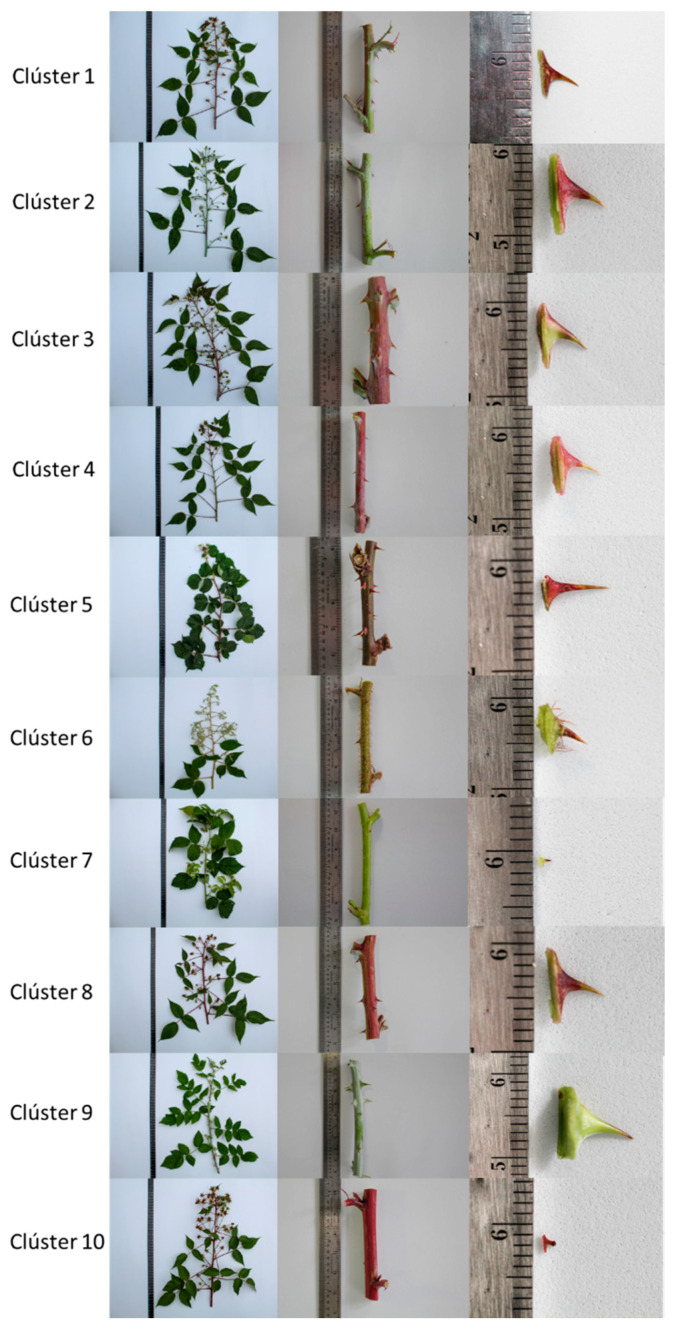
Morphological characteristics of the ten clusters formed by standardized K-means and fed with the variables prioritized by the optimized RF method.

**Table 1 plants-10-00247-t001:** Blackberry (*Rubus* spp.) accessions, selected from the blackberry germplasm bank for morphological characterization.

Specie	Unique Identification Code (CUI)	Specie	Unique Identification Code (CUI)
*Rubus* sp. (I)	72600104	*Rubus glaucus* (CN)	72600111
*Rubus* sp. (I)	72600101	*Rubus glaucus* (CN)	72600087
*Rubus glaucus* (CN)	72600121	*Rubus glaucus* (CN)	72600079
*Rubus glaucus* (CN)	72600015	*Rubus* sp. (I)	72600086
*Rubus glaucus* (CN)	72600027	*Rubus glaucus* (WN)	72600072
*Rubus glaucus* (CN)	72600028	*Rubus* sp. (CN)	72600090
*Rubus glaucus* (CN)	72600034	*Rubus glaucus* (CN)	72600068
*Rubus glaucus* (CN)	72600042	*Rubus* sp. (I)	72600115
*Rubus glaucus* (CN)	72600044	*Rubus glaucus* (CN)	72600106
*Rubus glaucus* (CN)	72600045	*Rubus glaucus* (CN)	72600116
*Rubus* sp. (WN)	72600108	*Rubus glaucus* (CN)	72600080
*Rubus glaucus* (CN)	72600050	*Rubus* sp. (I)	72600081
*Rubus glaucus* (CN)	72600094	*Rubus* sp. (I)	72600112

Cultivated natives (CN); introduced (I); wild native (WN).

**Table 2 plants-10-00247-t002:** Morphological variables used in the characterization of blackberry (*Rubus* spp.) accessions in this study.

Plant and Organs	Morphological Variables
Plant growth habit	(V7) Plant size (types of plant growth)
Stem	(V8, V9) Primary color of the stem surface
	(V10, V11) Secondary color of the stem surface
	(V12) Density of stem pubescence (tector hairs)
	(V13) Presence of trichomes on the stem (glandular hairs)
	(V14) Color of stem trichomes
	(V15) Number of trichomes on the stem [Area 0.5 cm2]
	(V16, V17, V18) Color of stem pubescence
	(V19) Stem waxiness
	(V20) Stem shape
	(V21, V22) Color of the base of the stinger on the stem
	(V23) Shape of the stinger on the stem
	(V24) Length of the base of the stinger on the stem [mm]
	(V25) Length of the stinger on the stem [mm]
	(V26) Number of stingers in the stem internode
Leaf	(V27) Length of the internode on the stem [mm]
	(V28) Stipule length (from leaf base and petiole) [mm]
	(V29) Stipule protrusion length on petiole [mm]
	(V30) Shape of stipules on petiole
	(V31) Blade shape
	(V32) Number of leaflets on the leaf
	(V33) Margin of terminal leaflet
	(V34) Shape of the base of the terminal leaflet
	(V35) Shape of terminal leaflet apex
	(V36) Terminal leaflet shape
	(V37) Pubescence of terminal leaflet (tector hairs)
	(V38, V39) Color of the terminal leaflet bundle
	(V40, V41) Color of the underside of the terminal leaflet
	(V42) Length of terminal leaflet [mm]
	(V43) Terminal leaflet width [mm]
	(V44) Petiole length at terminal leaflet [mm]
	(V45) Petiole length in terminal leaflet [mm]
Flower	(V46) Number of stingers in the terminal leaflet
	(V47) Type of inflorescence
	(V48) Petal shape
	(V49) Petal length [mm]
	(V50) Petal width [mm]
	(V51) Color distribution on the petals (pigmentation of certain areas)

**Table 3 plants-10-00247-t003:** Ability of machine learning methods to select the importance and discriminant blackberry (*Rubus* spp.) accessions based on morphology descriptors.

Algorithm Type	Train Accuracy ^1^	Train Missclass Error ^1^	Validation Accuracy ^1^	Validation Missclass Error ^1^
Random Forest with numerical variables	0.612	0.387	0.643	0.356
Random Forest with all variables	0.734	0.265	0.768	0.231
Support Vector Machine linear	Na	Na	0.0903	0.906
Support Vector Machine radial	Na	Na	0.218	0.781
Neuronal Networks	Na	Na	0.312	0.687

^1^ Determined parameter using area under receiver operating characteristic (ROC) curve (AUC).

**Table 4 plants-10-00247-t004:** Statistics test associated with evaluation of clustering methods to discriminate blackberry (*Rubus* spp.) accessions based on morphology descriptors.

Clustering Method type	NVI	ARI	IS	CH	EN	PG
HAC (not standardized)	0.83189	0.06594	11.11800	147.1061	1.45703	0.45055
K-means (not standardized)	0.64981	0.19419	7.72195	103.6778	1.92456	0.31012
HAC (standardized)	0.64030	0.18111	8.03727	65.75058	1.96550	0.30829
K-means (standardized)	0.32366	0.45414	7.55724	24.58237	2.18674	0.12498
HAC (standardized and with reduced variables based on random Forest process)	0.65026	0.17933	1.36298	69.08629	1.94982	0.40440
K-means (standardized and with reduced variables based on random Forest process)	0.32122	0.46730	1.05589	20.30752	2.19465	0.14359

Normalized Variation Index (NVI); Adjusted Rand Index (ARI); Separation index (IS); Calinski-Harabasz Index (CH); Entropy (EN); Pearson Gamma (PG). HC: Hierarchical agglomerative clustering.

## Data Availability

Data associated with morphological characterization of *Rubus* genus germplasms are part of the country’s genetic resource; the nation’s protection laws do not allow the publication of specific results without prior authorization. In addition, the R code generated during the current study is available from the corresponding author on reasonable request. For the future as a group, we are working on the development of an R package and a jupyter notebook for Python.

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
