# Peer review of "Machine Learning Applications and Optimization of Clustering Methods Improve the Selection of Descriptors in Blackberry Germplasm Banks"

_plants, 2021, doi:10.3390/plants10020247_

Round 1

Reviewer 1 Report

The presented study takes a high promising approach to the issue of selecting suitable descriptors for the morphological differentiation of blackberries.
Starting from my professional focus, which is not entirely in the field of machine learning or artificial intelligence, I have noticed some points that can be optimised:

  1. The comprehensibility of the work would gain by a changed structure, especially the materials and methods should be explained at an early stage.
  2. In preliminary considerations, scientifically relevant aspects, such as environmental dependency and interdependencies of morphological variables should be presented with regard to their influence on the applicability of the algorithms.
  3. The conclusions should take up the objectives of the work outlined in the introduction in more detail and make a more differentiated statement on applicability.
  4. It is not apparent that the results of the study have been questioned with regard to botanical aspects.
  5. The morphological variables found by the RF algorithms show a strong tendency towards ratio-scaled features, which is not explained.
  6. It is not clear how extensive the database used is and how training and validation sets were formed. The underlying dataset should be provided as supplementary data.
  7. Figure 7 is described, but is not included

Author Response

This letter serves to inform the corrections of manuscript with code “973515.”, by myself and my colleagues, for consideration for publication in the important journal plants were successfully realized

We found the reviewer to be helpful, and we have responded to each of their comments in detail. Based on the reviewer comments, we have made modifications to the original manuscript and proofread it carefully.

We are very grateful to the reviewer, as their comments were very successful. We detail our responses to each of the comments below, and we believe that the manuscript has been highly improved. We hope that this contribution will now prove acceptable for publication in your journal. We refer to line numbers. In addition, the language of the document thoroughly reviewed and corrected, process made by a native in English language.

Reviewer 1

1. The comprehensibility of the work would gain by a changed structure, especially the materials and methods should be explained at an early stage.

R: Done. We agree with the reviewer, although the journal suggests this order, we decided to follow the recommendation and materials and methods section were moved after the introduction

2. In preliminary considerations, scientifically relevant aspects, such as environmental dependency and interdependencies of morphological variables should be presented with regard to their influence on the applicability of the algorithms.

R. Done. We agree with reviewer and we included a better description of all aspects as suggested. The selected descriptors are genetically stable and not influenced by the environment condition. In addition, the results of the optimization processes based on machine learning methods were applied on the replicas evaluated in the field in order to avoid the effect of artifacts such as the influence of the environmental condition in the differential expression of morphological markers. Please see lines 82 to 90; 116 to 119; 113 to 139 and 229 to 233.

3. The conclusions should take up the objectives of the work outlined in the introduction in more detail and make a more differentiated statement on applicability.

R. Done. We agree with reviewer and we included a better description of all aspects as suggested.

4. It is not apparent that the results of the study have been questioned with regard to botanical aspects.

R. Done. We agree with reviewer and we included a better description of all aspects as suggested. We try to emphasize and explain selected variables and their relationship with botanical and taxonomic aspects of the Rubus genus. Please see lines 366 to 368 and 386 to 389.

5. The morphological variables found by the RF algorithms show a strong tendency towards ratio-scaled features, which is not explained.

R. Done. We agree with reviewer and we included a better description of all aspects as suggested. We try to explain more clearly this concept. Please see lines 366 to 368.

6. It is not clear how extensive the database used is and how training and validation sets were formed. The underlying dataset should be provided as supplementary data.

R. Done. We understand the reviewer's concern. The data matrix was 55 columns (morphological descriptors) per row (accessions), where for each accession there were 3 replicas. We add many detail, but since these data are part of the country's genetic resource, the nation's protection laws do not allow the publication of specific results without prior authorization, for which reason we do not choose to present and publish the data matrix. Please see lines 159 to 162.

 7. Figure 7 is described, but is not included

R. Done. The Figure 7 was added please see lines 428 to 429.

Reviewer 2 Report

This paper is a wonderfully focused and well written evaluation of various unsupervised analytical methods for discriminating 26 blackberry accessions.  The short review is that it is a hammer looking for a nail.  The long review realizes its technical merits for applying these methods for improved germplasm curation and its ongoing effort to develop efficiencies in partitioning and identifying useful variation.

However thorough this evaluation of discriminant methods is, the limitation is how does one evaluate it based on an objective assessment of diversity among these accessions.  What is lacking i here i an otherwise laudable and technically thorough study, is the biology.  This is pretty important to the readers of this journal.  My feeling is that there should have been an independent way of assessing the AI functions based on an on objective assessment of the variation among the accessions using genotypic evidence such as relatedness.

My worry is that any machine learning implementation for a discriminant function will find some solution weather its biologically relevant or not...like looking for differences between leaves from the same tree.  In this case the reader isn't given enough information about the accessions and their taxonomic status to evaluate the mathematical results of AI.  While machine learning hold enormous promise for big data, this study doesn't adequately highlight it potential in such a small related set of accessions.

One suggestion is that the authors submit this to a journal more focused on the comparison of methods. Another suggestion is to highlight to potential this method offers for the biological questions the readership is more interested in.  Perhaps providing more information about the identity and relationships among the accessions in question, the heritability of these descriptors and their utility in curation would be a step in the right direction.

Author Response

This letter serves to inform the corrections of manuscript with code “973515.”, by myself and my colleagues, for consideration for publication in the important journal plants were successfully realized

We found the reviewer to be helpful, and we have responded to each of their comments in detail. Based on the reviewer’s comments, we have made modifications to the original manuscript and proofread it carefully.

We are very grateful to the reviewer, as their comments were very successful. We detail our responses to each of the comments below, and we believe that the manuscript has been highly improved. We hope that this contribution will now prove acceptable for publication in your journal. We refer to line numbers. In addition, the language of the document thoroughly reviewed and corrected, process made by a native in English language.

  1. This paper is a wonderfully focused and well written evaluation of various unsupervised analytical methods for discriminating 26 blackberry accessions.  The short review is that it is a hammer looking for a nail. The long review realizes its technical merits for applying these methods for improved germplasm curation and its ongoing effort to develop efficiencies in partitioning and identifying useful variation.
  2. Thanks you. The idea of our group is applied this approach in other germplasm bank and had a tools to use in quickly and accurately characterization of the biodiversity.
  3. However thorough this evaluation of discriminant methods is, the limitation is how does one evaluate it based on an objective assessment of diversity among these accessions.  What is lacking i here i an otherwise laudable and technically thorough study, is the biology.  This is pretty important to the readers of this journal.  My feeling is that there should have been an independent way of assessing the AI functions based on an on objective assessment of the variation among the accessions using genotypic evidence such as relatedness.
  4. Done. We agree with the reviewer and throughout the document we try to connect the ML approach used with biological, ecological and taxonomic meaning. We also highlight the need to search for alternatives tools to selection of morphological descriptors to improve the efficiency in the description and knowledge on genetic resources. Please see lines 82 to 90; 116 to 119; 113 to 149; 229 to 233; 359 to 378; 386 to 389; and 418 to 427.
  5. My worry is that any machine learning implementation for a discriminant function will find some solution weather its biologically relevant or not...like looking for differences between leaves from the same tree.  In this case the reader isn't given enough information about the accessions and their taxonomic status to evaluate the mathematical results of AI.  While machine learning hold enormous promise for big data, this study doesn't adequately highlight it potential in such a small related set of accessions

R: Done. We agree with the reviewer, and we included a better description of all aspects as suggested. We also clarify that the accessions with the greatest morphological variability within the germplasm bank were selected. Please see lines 162 to 165; 133 to 139; 359 to 378 and 419 to 428.

4.One suggestion is that the authors submit this to a journal more focused on the comparison of methods. Another suggestion is to highlight to potential this method offers for the biological questions the readership is more interested in.  Perhaps providing more information about the identity and relationships among the accessions in question, the heritability of these descriptors and their utility in curation would be a step in the right direction.

  1. Done. We agree with reviewer and we included a better description of all aspects as suggested. We try to emphasize and explain selected variables and their relationship with botanical, taxonomic and genetic aspects of the Rubus genus. Please see lines 133 to 139; 366 to 368 and 386 to 389. 
  2.  

Sincerely, on behalf of all authors,

Joaquin Guillermo Ramirez Gil

Ph.D. Assistant professor

Universidad Nacional de Colombia

Reviewer 3 Report

The paper needs to be improved to be considered for publication and I listed some of the notes to help you.
1- The Abstract does not describe the whole paper. Two missing issues, which are not obviously declared in the Abstract, are the results and the data that used.

2- Work challenges and contributions should be clear enough for readers. I would recommend making them as items bullet points.

3- Although the paper presented a good use of Machine learning, the novelty of the paper is weak. what is the difference between your work and the use of deep learning? what are the reasons to not employ it? have you updated the machine learning classifiers that used with a new idea?

4- The paper does not present well. Readers expect to read the Materials and Methods after the introduction to understand the concept. Table 4. should be presented before the results, not at the end.
5- If possible, add a related work section after the introduction and show the research gap of prior work.

6- in the introduction section, references are old, try to enrich your paper from 2020 references. same thing if you add a related work section.
I would suggest these references which are related to agriculture:
a-https://www.mdpi.com/2223-7747/8/11/468
b- https://www.mdpi.com/2223-7747/9/10/1302
c-https://www.mdpi.com/2223-7747/9/10/1319

7- In the results section, what is the equation of the evaluation methods. Accuracy itself is not a reliable measure. suggest measuring Precision, Recall, F1-score.

8- Figures 1 and 2 are not clear enough to read.

9- The conclusion is short which not describing the whole paper.

Author Response

This letter serves to inform the corrections of manuscript with code “973515.”, by myself and my colleagues, for consideration for publication in the important journal plants were successfully realized

We found the reviewer to be helpful, and we have responded to each of their comments in detail. Based on the reviewer comments, we have made modifications to the original manuscript and proofread it carefully.

We are very grateful to the reviewer, as their comments were very successful. We detail our responses to each of the comments below, and we believe that the manuscript has been highly improved. We hope that this contribution will now prove acceptable for publication in your journal. We refer to line numbers. In addition, the language of the document thoroughly reviewed and corrected, process made by a native in English language.

  1. The Abstract does not describe the whole paper. Two missing issues, which are not obviously declared in the Abstract, are the results and the data that used.

R: done. We agree with reviewer and in this section we included a better description of all aspects as suggested. Please see lines 24 to 39

  1. Work challenges and contributions should be clear enough for readers. I would recommend making them as items bullet points.

R: done. We agree with reviewer and we included this aspects as suggested. Please see lines 53 to 58; 59 to 67 and 68 to 72.

  1. Although the paper presented a good use of Machine learning, the novelty of the paper is weak. what is the difference between your work and the use of deep learning? what are the reasons to not employ it? have you updated the machine learning classifiers that used with a new idea?

R.Done. We appreciate the reviewer concerns, and indeed this contribution represents something superficial of the usual context for this topic. Nonetheless, the contribution is relevant to the broader topic we aren’t expert deep learning and can’t speak more directly to that topic, but we offer the contribution as a view from a distinct toolset that we hope will offer some insights into a complex topic that has not been treated date under tropical conditions.

4. The paper does not present well. Readers expect to read the Materials and Methods after the introduction to understand the concept. Table 4. should be presented before the results, not at the end.

R. Done. We agree with the reviewer, although the journal suggests this order, we decided to follow the recommendation and materials and methods section were moved after the introduction

  1. If possible, add a related work section after the introduction and show the research gap of prior work.

R: done. We agree with reviewer and we included these aspects as suggested. Please see lines 53 to 58; 59 to 67 and 68 to 72.

6.  In the introduction section, references are old, try to enrich your paper from 2020 references. same thing if you add a related work section.
I would suggest these references which are related to agriculture:
a-https://www.mdpi.com/2223-7747/8/11/468
b- https://www.mdpi.com/2223-7747/9/10/1302
c-https://www.mdpi.com/2223-7747/9/10/1319

R. Done. We agree with reviewer and in the introduction, we included a better description of all aspects as suggested. Please see lines 53 to 58; 59 to 67 and 68 to 72.

  1. In the results section, what is the equation of the evaluation methods. Accuracy itself is not a reliable measure. suggest measuring Precision, Recall, F1-score.

R: Done. We agree with the reviewer that there are multiple metrics, but that the accuracy is still adequate for comparing results using different ML methods (Saleem et al., 2019; Huang et al., 2005 and Bradley et al., 1997). In this sense, a better description is added that it was calculated with area under receiver operating characteristic (ROC) curve (AUC). Please see lines 201 to 206.

  1. Figures 1 and 2 are not clear enough to read.

R: Done. We agree with reviewer and Figures 1 and 2 we included details and increase de size to a better description of all aspects as suggested.

9. The conclusion is short which not describing the whole paper.

R. Done. We agree with reviewer and we included a better description of all aspects as suggested. Please lines 431 to 440.

Round 2

Reviewer 1 Report

The publication has improved significantly in its comprehensibility. As a proof-of-concept for the application of deep learning algorithms for the described field of application it is appropriate even though the database is relatively small.

Author Response

Dear reviewer, as a working group we are very grateful for your excellent work. Without their valuable contributions, the manuscript would not have been improved.

Reviewer 2 Report

A distinct improvement over the original submission.

Details:

ln 64. not a sentence!

Ln 82 relies on the information...

Ln 86. agronomic parameters is a vague term and they are not necessarily heritable.

ln 87. Characterization describes the measurement of key botanical traits also know in the PGR research as descriptors.  There is no guarantee that these are not plastic or influenced by environmental conditions.

ln 116. A complex idea and garbled sentence.

Ln 138. These reference don't support the conclusions about phenotypic stability...pare this back a bit.

Ln 160. replicates

Ln 359. This is a long run-on sentence that totally confuses the essential point...the variables were variable and informative.

Ln. 370.  Accessions are not represented by species...its the other way around...species are represented by accessions...

Author Response

Dear reviewer, as a working group we are very grateful for your excellent work. Without their valuable contributions, the manuscript would not have been improved.

All suggestion were successfully incorporated into the manuscript. 

Reviewer 3 Report

The paper has been improved in a good way, I highly recommend to cooperate with a deep learning specialist and work on your task with deep learning for future work. 

Author Response

(The authors gave the same response as above.)
